# Quick Estimation Model for Mapping Earthquake Impacts in Bogotá, Colombia

Hiroyuki Miura [1,*], Masashi Matsuoka [2], Juan C. Reyes [3], Nelson Pulido [4], Mitsufumi Hashimoto [5], Andrea C. Riaño [3], Alvaro Hurtado [3], Raul Rincon [6], Helber García [7] and Carlos Lozano [8]

1 Graduate School of Advanced Science and Engineering, Hiroshima University, Hiroshima 739-8527, Japan
2 Department of Architecture and Building Engineering, Tokyo Institute of Technology, Yokohama 226-8501, Japan; matsuoka.m.ab@m.titech.ac.jp
3 Department of Civil and Environmental Engineering, Universidad de los Andes, Bogotá DC 11711, Colombia; jureyes@uniandes.edu.co (J.C.R.); ac.riano384@uniandes.edu.co (A.C.R.); ai.hurtado77@uniandes.edu.co (A.H.)
4 National Research Institute for Earth Science and Disaster Resilience, Ibaraki 305-0006, Japan; nelson@bosai.go.jp
5 Kozo Keikaku Engineering Inc., Tokyo 164-0001, Japan; mitsu@kke.co.jp
6 Department of Civil and Environmental Engineering, Rice University, Houston, TX 77005, USA; raul.rincon@rice.edu
7 Servicio Geológico Colombiano (SGC), Bogotá DC 111321, Colombia; hgarcia@sgc.gov.co
8 Instituto Distrital de Gestión de Riesgos y Cambio Climático (IDIGER), Diagonal 47 Number 77A-09 Int. 11, Bogotá DC 111071, Colombia; clozano@idiger.gov.co
* Correspondence: hmiura@hiroshima-u.ac.jp

**Abstract:** Early disaster responses in damaged areas after a large earthquake are indispensable for stakeholders to assess and grasp the impacts such as building and infrastructure damage and disrupted community functionality as soon as possible. This study introduces a quick estimation model for mapping seismic intensities and building losses in Bogotá, the capital city of Colombia. The model uses ground motion records in the seismic network, soil maps of average shear-wave velocity in the upper 30 m (Vs30) with site amplifications, building inventory, and vulnerability functions for all building types. The spatial distribution of ground motion intensities, including spectral accelerations, was estimated by interpolating the observed seismic intensities with the Vs30-based site amplifications. The losses (repair cost) for all the buildings were evaluated by integrating the estimated spectral accelerations, the building inventory, and the vulnerability functions. The spatial distributions of seismic intensities and building losses can be computed within a few minutes immediately after triggering earthquake motions in the seismic network. The proposed model demonstrates evaluations of the impacts for the Mw6.0 earthquake that occurred on December 2019 and an earthquake scenario with Mw7.0 from an active fault near the Bogotá region.

**Keywords:** seismic intensity; building loss; vulnerability; Bogotá

## 1. Introduction

Identification of the spatial distribution of seismic intensities and building damage immediately after large earthquakes is crucial for the administrators of the national and local governments to understand the impacts and to define emergency response strategies. For this purpose, the ShakeMap system has been developed to rapidly and automatically generate seismic intensity maps after earthquakes by spatially interpolating observed ground motion intensities [1–3]. Similarly, early ground motion estimation systems have been implemented in some countries and cities, such as Japan and European countries [4–9].

Estimation of building damage distribution is important because of the human losses and the economic impacts that structural damage and failure can produce. Building damage and/or loss maps have been implemented in some seismic monitoring systems [4,7,8,10].

Up-to-date building inventory data and fragility/vulnerability functions required for damage mapping have been developed in each region for predicting building damage due to real and scenario earthquakes [11–17]. Such seismic shaking and building damage mapping systems, however, require a real-time seismic monitoring network, detailed soil amplification data, building inventory data, and fragility/vulnerability functions suitable for each region. The number of countries and cities where such quick estimation systems have been implemented is still limited because of the huge costs and investments needed for preparing the input data.

Bogotá, the capital city of Colombia, is one of the largest metropolitan cities in Latin America with a population of approximately eight million and a total number of buildings of over 800,000. Bogotá has been affected by magnitude 6–7 class crustal earthquakes triggered by active faults in the Eastern Andes range at intervals of approximately 100 years [18]. The largest event in this region assessed from the historical earthquake records was the Ms 7.1 in August 1917 [18]. Such a large earthquake has not been recorded near Bogotá since 1917 when Bogotá experienced the most damaging earthquake to date. Therefore, we should consider countermeasures against destructive magnitude 7 class earthquakes that will occur in the near future.

In order to estimate damage information for emergency responses in earthquake crises, the Laboratory of Automatic Seismic Instrumentation (LISA, Laboratorio de Instrumentación Sísmica Automática) has been developed in Bogotá [19,20]. The LISA produces shakemaps and building damage maps with block-level resolution including economic losses of the damaged buildings. The typical size of the blocks in Bogotá was several thousand to tens of thousands of square meters. The obtained shakemaps and damage maps by LISA can be sent to designated city officials by email and short message service (SMS) within five minutes after the earthquake. Although the system is a convenient and valuable tool to immediately assess the spatial distributions of ground shakings and building damage, the LISA still has limitations and uncertainties, especially in estimating shakemaps as follows:

- No seismic event information (such as the one provided by the National Seismic Network of Colombia [21]) was incorporated into the automatic shakemap calculations of LISA [20], hindering its timely and accurate release of information.
- The seismic motion intensities on the ground surface were estimated based on a single ground motion record observed at a bedrock site within Bogotá, and, therefore, the system may not properly reflect the large spatial variability expected in near-source ground motions, due to finite fault ruptures of potential M6-7 class earthquakes nearby.
- The geotechnical model used for detailed seismic response analysis of soils at a block level (~hundreds of meters), was interpolated by using a probabilistic approach from the geotechnical information available at 23 boreholes (including measurements of shear-wave velocity, Vs) within Bogotá (every ~8 km) [20].

Significant discrepancies might be found between the probabilistically determined models and more realistic models investigated by dense geophysical explorations. Since the Vs models strongly control the amplitudes of the seismic motions on the ground surface, a more realistic Vs model in Bogotá has been expected for accurate seismic hazard evaluations. Furthermore, the shakemaps in the LISA did not reflect the seismic intensities observed at other ground surface sites in Bogotá. More accurate shakemaps can be produced by adding a large number of ground motion records observed at multiple sites available in the seismic monitoring network currently in operation.

To address these issues, we have conducted very dense geophysical explorations (every ~1 km) based on single and array microtremor measurements as well as existing gravity measurements (~1 km), for developing more accurate three-dimensional (3D) Vs structures in the Bogotá basin [22,23]. We included empirical models of site amplification factors for peak ground accelerations (PGA), peak ground velocities (PGV), and spectral accelerations (Sa) that are dependent on the average shear-wave velocities in the upper 30 m (Vs30) obtained in geophysical explorations [24,25]. More site-specific and reliable

shakemaps can be created by using observed ground motion data at multiple sites and the developed site amplification factors. Based on the previous studies, this study introduces a quick estimation model for mapping earthquake impacts, such as shakemaps and building damage maps, including building economic losses, based on seismic observation data, the Vs30 map, and building vulnerabilities.

## 2. Development of Quick Estimation Model

### 2.1. Accelerograph Network in Bogotá

Figure 1 shows an overview of the developed quick ground motion and building loss estimation model for Bogotá. The model is based on ground motion records observed by the seismic network operated by The Instituto Distrital de Gestión de Riesgos y Cambio Climático (IDIGER, District Institute of Risk Management and Climate Change) in Bogotá named Red de Acelerógrafos de Bogotá (RAB) [26]. Seismic observations at the ground surface in Bogotá have been operated by IDIGER at 29 sites, as shown in Figure 2. The strong motion seismographs named BASALT, OBSIDIAN, K2, ETNA-2, and ETNA in Kinemetrics Inc. have been used by the IDIGER in the seismic network (see Table 1). At the 25 stations where the BASALT, OBSIDIAN, and ETNA-2 equipment are installed, the observed seismic data are transmitted to the IDIGER headquarters by broadband telemetry system immediately after triggering a tremor. Peak ground accelerations (PGA), peak ground velocities (PGV), and acceleration response spectra (Sa) are calculated from the transmitted records, and they are used in the following quick estimations.

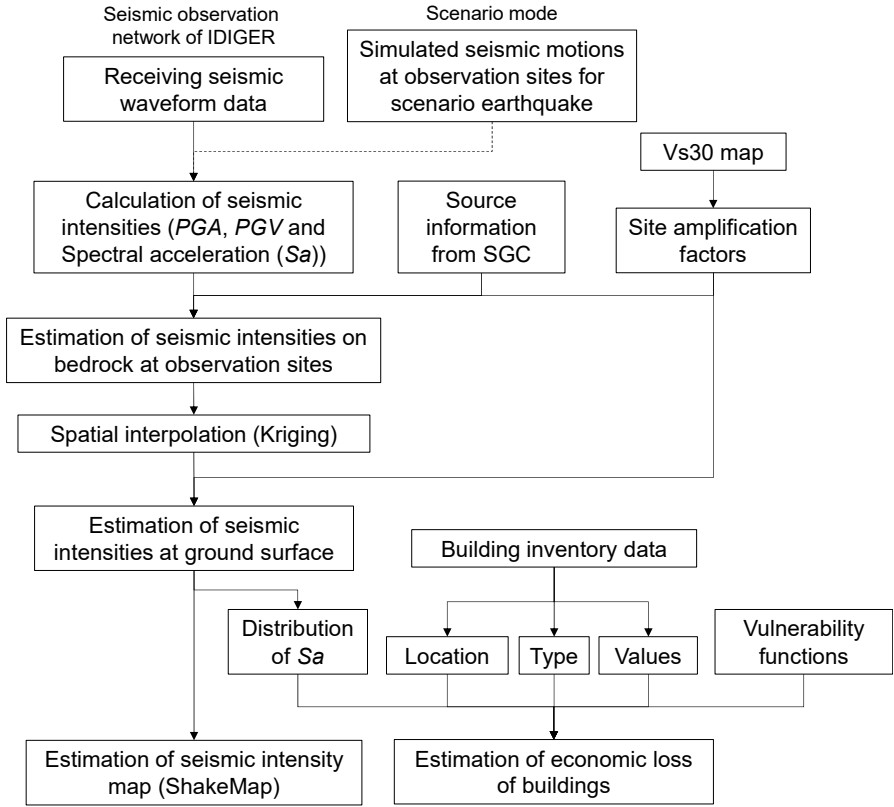

**Figure 1.** Diagram for quick estimation model for shakemap and economic loss of buildings developed for Bogotá, Colombia.

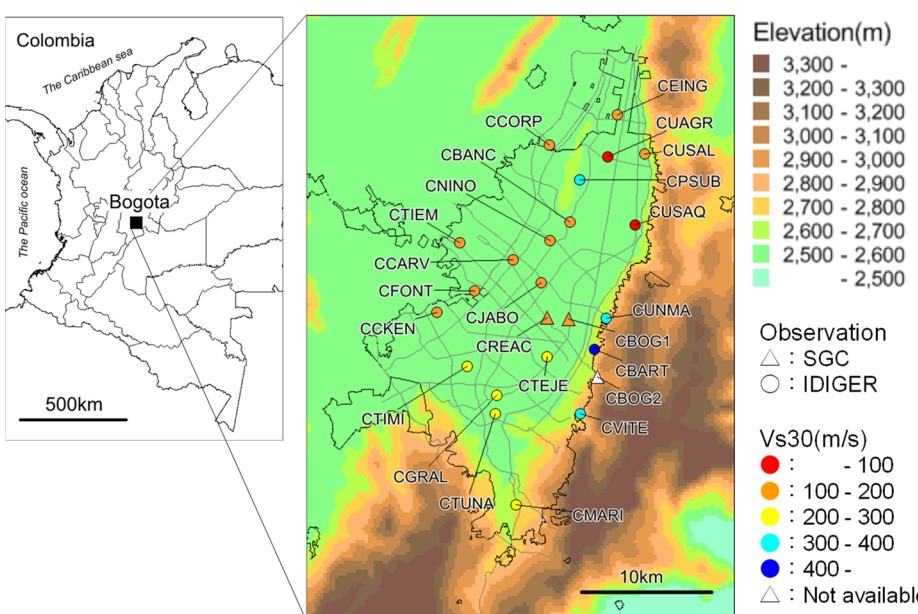

**Figure 2.** Seismic ground observation network by IDIGER in Bogotá with their Vs30 values. The map shows the distribution of the observation stations with topographical map.

**Table 1.** Seismic motion observation sites by IDIGER in Bogotá.

| ID | Site | Longitude (deg.) | Latitude (deg.) | Elevation (m) | Vs30 (m/s) | Type of Seismograph |
|---|---|---|---|---|---|---|
| 1 | CARAN | −74.1128 | 4.6456 | 2555 | 161 | ETNA-2 |
| 2 | CARTI | −74.1234 | 4.5469 | 2569 | 333 | BASALT |
| 3 | CBANC | −74.0790 | 4.7085 | 2552 | 123 | ETNA-2 |
| 4 | CBART | −74.0618 | 4.6200 | 2671 | 425 | BASALT |
| 5 | CBOSA | −74.1922 | 4.6065 | 2552 | 220 | BASALT |
| 6 | CCARV | −74.1188 | 4.6823 | 2556 | 120 | BASALT |
| 7 | CCKEN | −74.1724 | 4.6459 | 2548 | 173 | BASALT |
| 8 | CCORP | −74.0940 | 4.7619 | 2554 | 98 | ETNA-2 |
| 9 | CDIOS | −74.0884 | 4.5899 | 2583 | - | ETNA-2 |
| 10 | CEING | −74.0460 | 4.7836 | 2562 | 97 | ETNA-2 |
| 11 | CFLOD | −74.1464 | 4.7297 | 2557 | 112 | ETNA |
| 12 | CFONT | −74.1456 | 4.6609 | 2546 | 157 | BASALT |
| 13 | CGRAL | −74.1301 | 4.5879 | 2566 | 255 | OBSIDIAN |
| 14 | CJABO | −74.0993 | 4.6665 | 2554 | 100 | ETNA-2 |
| 15 | CLAGO | −74.1003 | 4.7180 | 2552 | - | ETNA-2 |
| 16 | CMARI | −74.1171 | 4.5120 | 2689 | 257 | BASALT |
| 17 | CNIÑO | −74.0931 | 4.6962 | 2555 | 109 | ETNA-2 |
| 18 | CPSUB | −74.0726 | 4.7379 | 2588 | 139 | BASALT |
| 19 | CSMOR | −74.1701 | 4.5746 | 2783 | 398 | BASALT |
| 20 | CTEJE | −74.0951 | 4.6146 | 2566 | 214 | ETNA-2 |
| 21 | CTIEM | −74.1560 | 4.6944 | 2552 | 123 | ETNA |
| 22 | CTIMI | −74.1510 | 4.6083 | 2559 | 186 | BASALT |
| 23 | CTUNA | −74.1311 | 4.5752 | 2563 | 257 | BASALT |

**Table 1.** *Cont.*

| ID | Site | Longitude (deg.) | Latitude (deg.) | Elevation (m) | Vs30 (m/s) | Type of Seismograph |
|----|------|------------------|-----------------|---------------|------------|---------------------|
| 24 | CTVCA | −74.0847 | 4.7179 | 2652 | 142 | BASALT |
| 25 | CUAGR | −74.0527 | 4.7541 | 2561 | 93 | ETNA |
| 26 | CUNMA | −74.0539 | 4.6416 | 2679 | 333 | ETNA |
| 27 | CUSAL | −74.0267 | 4.7558 | 2567 | 114 | ETNA-2 |
| 28 | CUSAQ | −74.0339 | 4.7062 | 2565 | 100 | ETNA-2 |
| 29 | CVITE | −74.0717 | 4.5752 | 2777 | 555 | ETNA-2 |

The Servicio Geológico Colombiano (SGC, Colombian Geological Survey) has developed a nationwide seismic network (RSNC) for monitoring activities of earthquakes and volcanos since 1993 [21,27,28]. When a tremor is detected by the seismic network, the source information, such as location, magnitude, and depth estimated from the observed waveforms, is immediately distributed to registered people and organizations by email service. The obtained seismic data in IDIGER are linked with the source information provided by email service from the RSNC in the model based on the registered GPS clock.

### 2.2. Vs30 Map and Site Amplification Factors

Miura et al. [24] found that long-period (approximately 2 s) surface waves were significantly predominant in ground motions observed within the Bogotá basin in cases of shallow earthquakes due to the basin-edge effect of the deep sedimentary basin. On the contrary, body waves directly propagated from the earthquake source were dominant in deep earthquakes. These authors also found strong correlations between Vs30s and site amplification factors and proposed a Vs30-dependent site amplification factor model of Sa, PGA, and PGV for surface wave type and body wave type [24,25]. Figure 3 shows the Vs30-dependent site amplification factors of Sa for surface wave and body wave types for several Vs30s proposed in reference [24]. It must be mentioned that the seismic records used had PGA values smaller than 80 cm/s$^2$, thus strong shaking amplification effects, such as nonlinear ground responses, still require further attention.

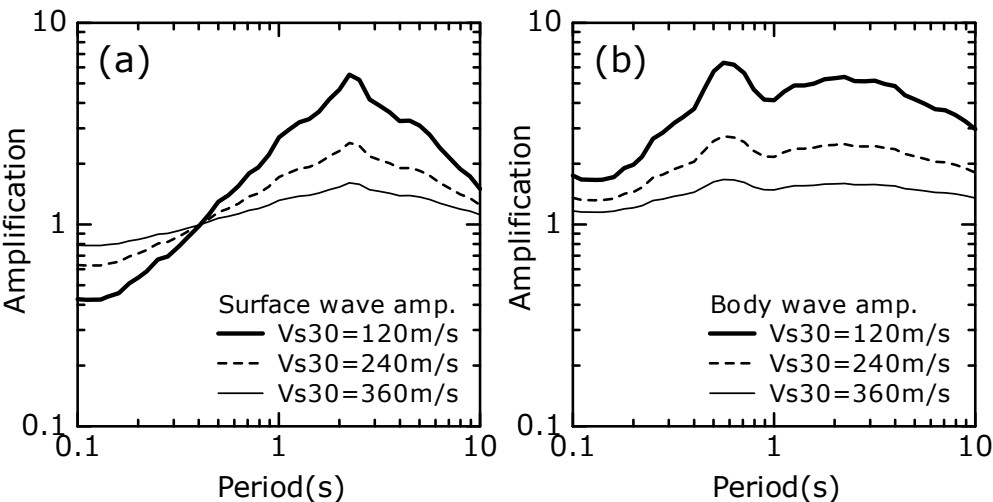

**Figure 3.** Vs30-dependent site amplification factors for (**a**) surface waves and (**b**) body waves in Bogotá proposed in Miura et al. [24].

Although Vs30s represent characteristics of very shallow surface soil, Miura et al. [24] observed that the ground motions in Bogotá can be characterized by the Vs30s, and the developed Vs30-based site amplification factors accurately reproduced the ground motion characteristics observed in Bogotá's basin. Given this observation, we used the Vs30 model

in [24] as a proxy of site amplification factors on the quick loss estimation model. The Vs30 map included in the system uses data from three-dimensional (3D) velocity structures developed for the region [22]. For the development of the 3D model, array measurements and single-site observations of microtremors were performed at approximately 300 and 800 sites, respectively, in and around the city of Bogotá and 800 sites in the surrounding areas [23,24]. The shear-wave velocity profiles at the observation sites were estimated by inversion analysis of the microtremor-derived phase velocities. In addition, the 3D material model was constructed to characterize properties up to the seismic bedrock with a shear-wave velocity of approximately 3000 m/s. Figure 4 shows the Vs30 map with a spatial resolution of 250 m in Bogotá. Since Bogotá is located on a large basin with a long sedimentation process of lacustrine fluvio-glacial deposits, the surface soil condition is very soft. As shown in Figure 4, the Vs30 values in most of the city are smaller than 300 m/s, and small values of Vs30s with 200 m/s or less are widely distributed.

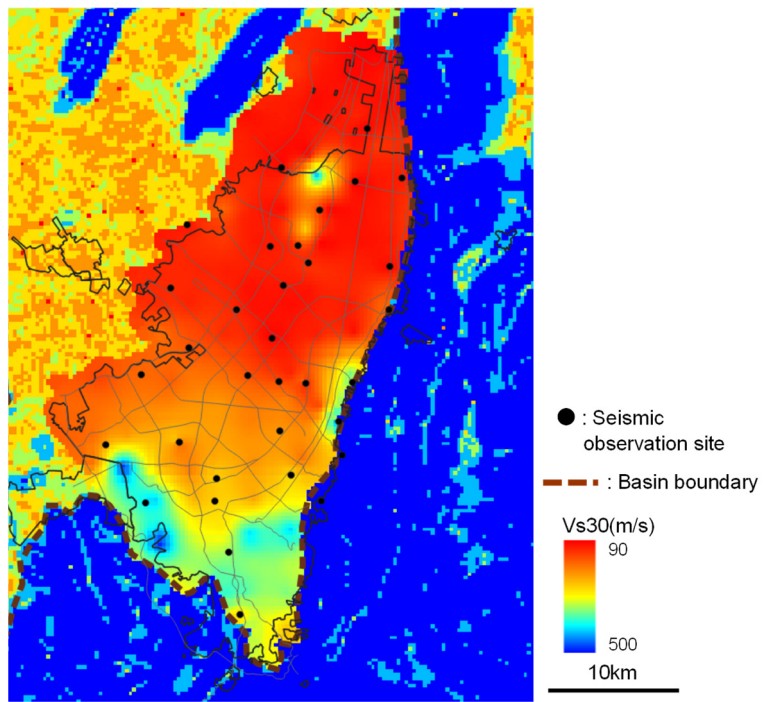

**Figure 4.** Vs30 map in Bogotá (after Pulido et al. [23]). Points and dotted line indicate the locations of seismic observation sites and the basin boundary, respectively.

Figure 5 shows the schematic diagram for estimating shakemaps from observation records and site amplification factors obtained from the Vs30 map. First, the seismic intensities at the bedrock of the observation stations are estimated by de-amplifying the seismic intensities observed at the surface with the site amplification factors. Then, the surface wave amplification factors are applied for shallow earthquakes (less than 150 km depth) whereas the body wave amplification factors are used for deeper earthquakes (larger than 150 km depth) based on the source information estimated by the RSNC. Next, the distributions of the seismic intensities on the bedrock are obtained by spatial interpolation based on the simple Kriging [29], since the technique was also adopted in a similar quick estimation model of strong ground motion maps for Japan [6]. The distributions of the seismic intensities at the surface are estimated from the spatially interpolated seismic intensities on the bedrock and the Vs30 map-based site amplification factors. With this, the system outputs the maps of PGA, PGV, and Sa for the period of 0.1 to 10.0 s as shakemaps.

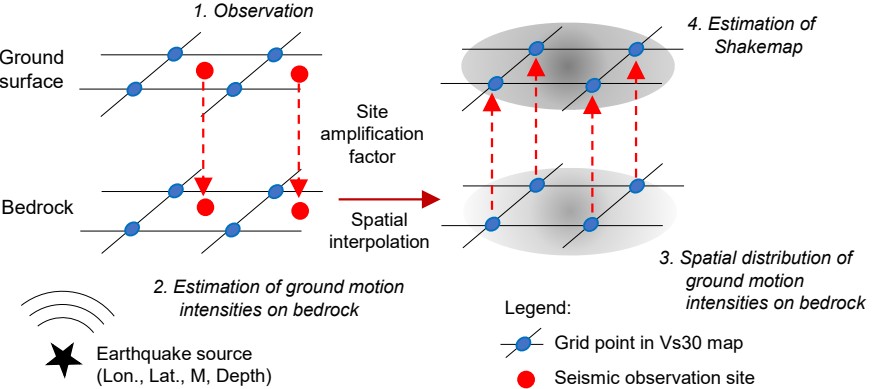

**Figure 5.** Schematic diagram for estimation of shakemap based on observation seismic data and site amplification factors. Step 1 to 4 in the figure indicates procedures for observations and estimations of ground motion intensities at ground surface and bedrock.

Note that explicit nonlinear soil behavior expected during strong shaking, in the rare case of a large earthquake near Bogotá, was not considered in the shakemaps because the developed amplification models were based on seismic records whose PGA is smaller than 80 cm/s$^2$. However, the observed strong motion records that possibly experienced nonlinearity will be also included for the shakemap calculations, and, therefore, the effect of nonlinearity in ground motions would be incorporated to a first order. Anyway, this limitation may be addressed in the future by extending the amplification models with analytical strong motion approaches.

### 2.3. Building Inventory and Vulnerability Functions

Building inventory databases and building fragility/vulnerability functions are required for estimating citywide building damage [30]. Detailed databases are available in Bogotá as part of the "Spatial Data Infrastructure of the Capital District" [31]; these databases consist of cadastral geographic information layers at 'lot-level' and 'construction-level'. The lot-level inventory data includes lot area and corresponding social-economic class. The number of lots in the existing database is approximately 890,000. Additionally, the construction-level data characterizes the building's footprints and are modeled by 1.6 million polygons, which include data about the number of stories and the information required to define the buildings' use. These existing databases do not include building typology that is the most important factor for evaluating seismic capacity of buildings; hence, a structural typology assignation model was used for this purpose.

A matrix-like algorithm was used to assign the building typologies in the exposure inventory. The algorithm assigns the most likely structural typology based on the building use, number of floors, socioeconomic level, and slope of the terrain. All possible combinations were considered in the assignation model. These characteristics were found to create spatially distributed clusters in dense cities, which relate well with the building typology clusters. For example, high-rise buildings with commercial usage (e.g., offices) located on high socio-economic levels are commonly found in the financial sector, which usually are reinforced concrete dual-systems; likewise, unreinforced, and confined masonry are observed to be dominant typologies in residential low-rise buildings located in low-income areas (probably with moderate-to-high slope terrain). The algorithm was developed in previous studies at the Universidad de los Andes (Uniandes), Colombia, using cadastral databases, web-based inspection of thousands of buildings, and in-person validation for hundreds of buildings and adapted to the current study's particularities [32]. Table 2 shows the building typologies in the developed inventory data. The building typologies from the tree-like model resulted in 23 classes. Structural types were classified as Adobe, MSC (unreinforced masonry bearing walls), MR (reinforced masonry), PCRD (concrete moment frame), PCRM (concrete frame with unreinforced masonry infill walls), and others. The

design level (low, moderate, and high) was assigned to the PCRD and PCRM. The distribution of the classified structural types is illustrated in Figure 6a. The structural period in Table 2 indicate the fundamental vibration period for each building typology estimated from the typical height and the structural type [16,17,33]. The number of buildings in each class is also shown in Table 2. Approximately 90% of the buildings are low-rise dwellings with 1–3 stories, and the MSC type is predominant in Bogotá city. Figure 6b shows the distribution of the building heights; high-rise buildings are observed in the northeast part of the city.

**Table 2.** Building typologies and their characteristics in building inventory data of Bogotá, Colombia.

| No. | Structural Code | Description | Height | Stories | Typical Height (m) | Structural Period (s) | Design Level | Number | Percentage (%) |
|---|---|---|---|---|---|---|---|---|---|
| 1 | ADOBE | Adobe | Low | 1–2 | 6.1 | 0.50 | - | 19,193 | 1.20 |
| 2 | MSC1_3 | Unreinforced masonry bearing walls | Low | 1–3 | 4.6 | 0.35 | - | 1,186,283 | 73.97 |
| 3 | MSC4_5 | | Mid | 4–5 | 10.7 | 0.50 | - | 26,913 | 1.68 |
| 4 | MR1_3 | Reinforced masonry bearing walls | Low | 1–3 | 6.1 | 0.35 | - | 206,205 | 12.86 |
| 5 | MR4_5 | | Mid | 4–5 | 15.2 | 0.56 | - | 15,645 | 0.95 |
| 6 | PCRDMI1_3 | Concrete moment frames | Low | 1–3 | 6.1 | 0.40 | Low | 253 | 0.02 |
| 7 | PCRDMO1_3 | | Low | 1–3 | 6.1 | 0.40 | Moderate | 14,943 | 0.93 |
| 8 | PCRDMI4_5 | | Mid | 4–5 | 15.2 | 0.75 | Low | 64 | 0.00 |
| 9 | PCRDMO4_5 | | Mid | 4–5 | 15.2 | 1.45 | Moderate | 5190 | 0.32 |
| 10 | PCRDMO6_12 | | High | 6–12 | 36.6 | 0.56 | Moderate | 3308 | 0.21 |
| 11 | PCRM_DMI4_5 | Concrete frame with unreinforced masonry infill walls | Mid | 4–5 | 15.2 | 0.56 | Low | 10,299 | 0.64 |
| 12 | PCRM_DMO4_5 | | Mid | 4–5 | 15.2 | 0.56 | Moderate | 9904 | 0.62 |
| 13 | PCRM_DES4_5 | | Mid | 4–5 | 15.2 | 0.56 | High | 1390 | 0.09 |
| 14 | PCRM_DMI6_12 | | High | 6–12 | 36.6 | 1.09 | Low | 588 | 0.04 |
| 15 | PCRM_DMO6_12 | | High | 6–12 | 36.6 | 1.09 | Moderate | 4476 | 0.28 |
| 16 | PCRM_DES6_12 | | High | 6–12 | 36.6 | 1.09 | High | 1365 | 0.09 |
| 17 | PCRM_DMO12_20 | | High | 12–20 | 56.0 | 1.76 | Moderate | 490 | 0.03 |
| 18 | PrFC4_5 | Precast concrete tilt-up walls | Mid | 4–5 | 15.2 | 0.56 | - | 12,553 | 0.78 |
| 19 | SC6_12 | Reinforced concrete frames and concrete shear walls | High | 6–12 | 36.6 | 1.09 | - | 574 | 0.04 |
| 20 | SC_20 | | High | 12+ | 65.0 | 2.01 | - | 47 | 0.00 |
| 21 | PRT_CER_MUR | Reinforced concrete frames and steel truss girder (Warehouses) | - | All | 4.6 | 0.32 | - | 11,058 | 0.69 |
| 22 | BOD_PEQ | Steel light frame | - | All | 4.6 | 0.40 | - | 42,023 | 2.62 |
| 23 | BOD_GRN | | - | All | 4.6 | 0.40 | - | 30,948 | 1.93 |
| Total | | | | | | | | 1,603,712 | 100.00 |

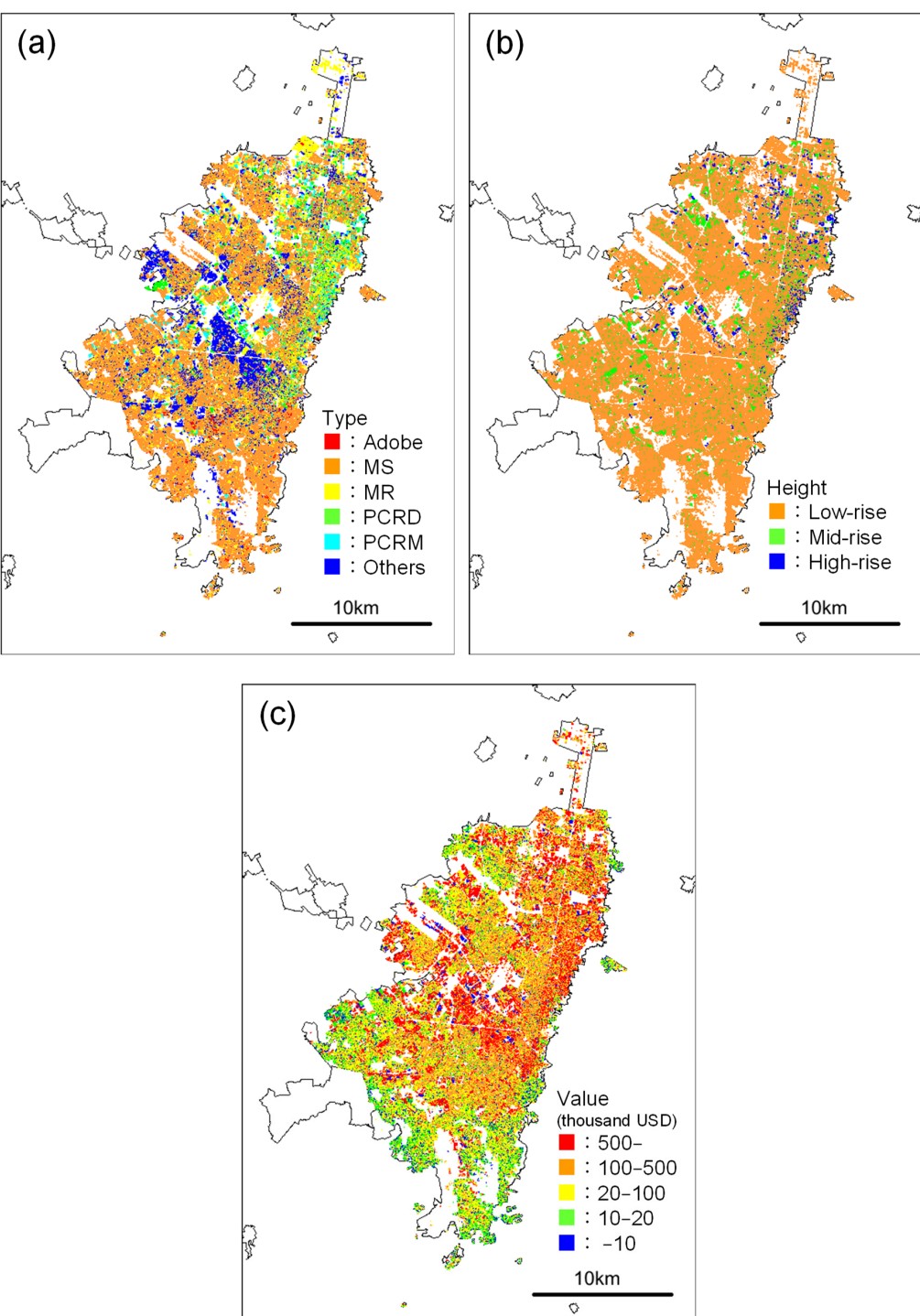

**Figure 6.** Building inventory data. (**a**) Building typologies (see Table 2), (**b**) Building heights, and (**c**) Building values in USD.

Another important characteristic that existing building databases do not contain is the building replacement cost. Considering that the replacement costs intent is to relate the building's damage to the expected economic losses, a cost estimation model is built into this study based on factors that capture the variation in the building's costs produced by phenomena such as predominant construction material, construction quality, construction age, variation in finishing costs, and quality, among other variables. Information that supports the abovementioned properties is not easy to collect, thus a construction price index (per square meter) is defined for different combinations of (1) building uses,

(2) socio-economic levels, (3) locality, and (4) building height, assuming that these factors jointly influence the buildings' replacement cost at Bogotá. The last factor was used only to differentiate low-to-midrise constructions from high-rise buildings. The price index model was constructed using available information from the public contracts databases [34], international construction surveys [35], commercial building costs databases [36], surveying the commercial costs of different building configurations in each of the 20 localities in Bogotá [36,37], and the use, structure-height, and socio-economic classification according to the cadastral database [31]. The building replacement cost was obtained by multiplying the corresponding price index per use by each constructed area minus the relative land value associated for each building into a lot unit. Here, it was assumed that the estimated building value corresponds to the replacement cost including materials, supplies, labors, tools, equipment, machines, and transportations and excludes the following costs: land acquisition, relocating utilities, construction technology equipment, and commercial aspects. The residential and commercial uses account for more than 90% of the regional exposed value, and the total building inventory cost was appraised at 720 billion COP (in 2019), a cost validated against the Real Estate Census (in 2019) reported in [38]. Figure 6c shows the obtained distribution of the building replacement costs in thousand USD under the currency exchange rate as of 2019 (1 USD = 3333.3 COP).

A unique vulnerability function was assigned to each building typology defined in the exposure database. The vulnerability functions represent the relationship between seismic intensity and loss value, which is expressed as total repair cost normalized by building replacement value (i.e., the mean damage ratio, MDR). The vulnerability functions were adapted from the previous studies [16,33], performing adjustments to represent the local characteristics of the building portfolio, as shown in Figure 7. The horizontal axis indicates spectral acceleration at the typical fundamental period of vibration of the building type (shown also in Table 2); the vertical axis corresponds to the MDR in percentage. More details on the exposure database and vulnerability curves for the city of Bogotá can be found in [33]. In this model, it was assumed that the MDRs in the vulnerability functions were zero when the MDRs were smaller than 0.1. This decision was based on the assumption that owners would not repair slight damage, such as hair cracks on non-structural walls.

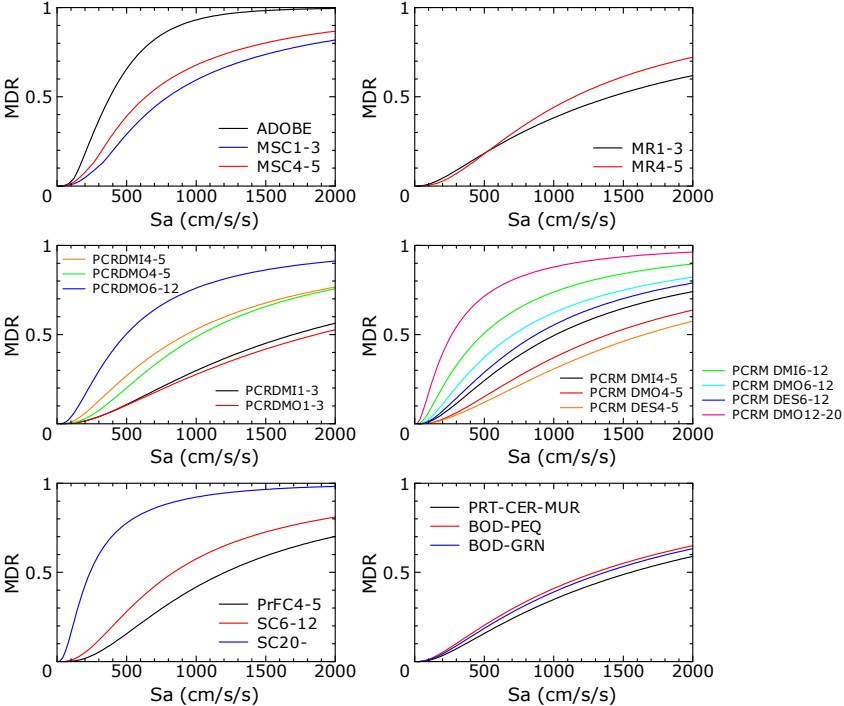

**Figure 7.** Vulnerability functions for building types in Bogotá.

*2.4. Shakemaps and Building Loss Maps*

The building loss is calculated first using the estimated spectral acceleration at ground level and the vulnerability curve to obtain the predicted MDR; then, the expected losses are computed as the building replacement cost multiplied by the MDR. The estimated building losses are aggregated by the 250 m mesh map, and the total loss in each mesh is illustrated into a building loss map.

The shakemaps and building loss maps can be obtained within several minutes after receiving seismic data transmitted from the observation stations. The city administrators and operators can confirm the estimation results displayed on internet browsers. The system automatically produces shakemaps and building loss maps in raw image file (RAW format) and compressed image file (PNG format) with the georeferencing information in HDR format. The color scales of the maps are automatically given according to the maximum and minimum values of the estimations.

## 3. Shakemaps for the Mesetas Earthquake on 24 December 2019

This chapter introduces an example for the estimation of shakemaps for a real earthquake. An earthquake with magnitude Mw6.0 occurred in the Mesetas district approximately 100 km south of Bogotá at 14:03 (local time) on 24 December 2019 [39]. The epicenter was located at a longitude of −74.18 degrees and a latitude of 3.46 degrees. The focal depth was about 13 km according to the source information by the SGC. This earthquake was the largest crustal earthquake in the last 20 years in Colombia [40]. The seismographs at 12 stations of the IDIGER seismic network were triggered in the earthquake.

Figure 8 shows the pasteups of the observed acceleration and velocity waveforms in Bogotá. Since the CMARI seismic station is located near the epicenter, large amplitudes were observed in accelerations and velocities. Large amplitudes were observed also at CUSAQ whereas the site is located far from the epicenter. As shown in Figure 4, the northeast area of the Bogotá basin around CUSAQ is covered with very low Vs30s, indicating that very soft soils with thick deposits exist in the area. Due to such very soft soils, large amplifications produced stronger ground shakings at the site. On the contrary, very small amplitudes were observed at CBART. Since CBART is located at the rock site of the eastern edge of the basin, very small amplifications were expected at the site. Figure 9 shows the acceleration response spectra at typical sites with their Vs30 values. As described above, smaller spectral accelerations were observed at CBART with the Vs30 of over 500 m/s. The responses increased as the Vs30s decreased. Especially, large amplitudes at approximately 1–2 s were observed at CUSAQ. Such strong shakings at longer periods confirm large amplifications due to soft soils, as shown in Figure 3. These results indicate that the site conditions strongly control the amplitudes of the ground shakings within the basin, and accurate soil models would be important for reliable shakemaps.

After receiving the observed waveform data, the developed system produced shakemaps and building loss maps within one minute. Although the building inventory data included more than 1.6 million, building damage and their repair costs were calculated with a very short computation time. The surface wave amplification models were automatically adopted in calculating the seismic intensities since the focal depth was only 13 km. Figure 10 shows the shakemaps of the estimated PGAs, PGVs, and spectral accelerations of typical periods for the earthquake. The southern areas show larger seismic intensities in the PGA map and the short-period spectral accelerations for the period of 0.35 s and 0.5 s because the source of the earthquake was closer. However, the northeast areas also show larger intensities especially in the PGV and longer period spectral accelerations, such as 1.0 s and 2.0 s, because of the strong amplification due to the soft soil conditions in the area. These shakemaps show the site conditions typically affected the seismic intensities, especially for longer periods.

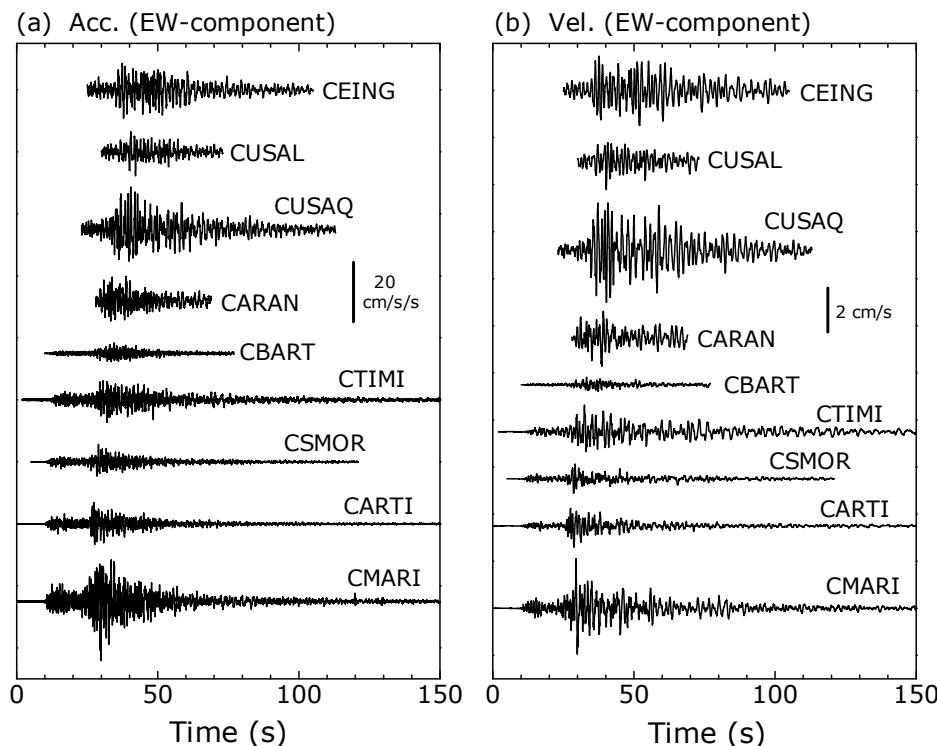

**Figure 8.** Observed time histories of EW component for (**a**) accelerations and (**b**) velocities in the 2019 Mesetas earthquake.

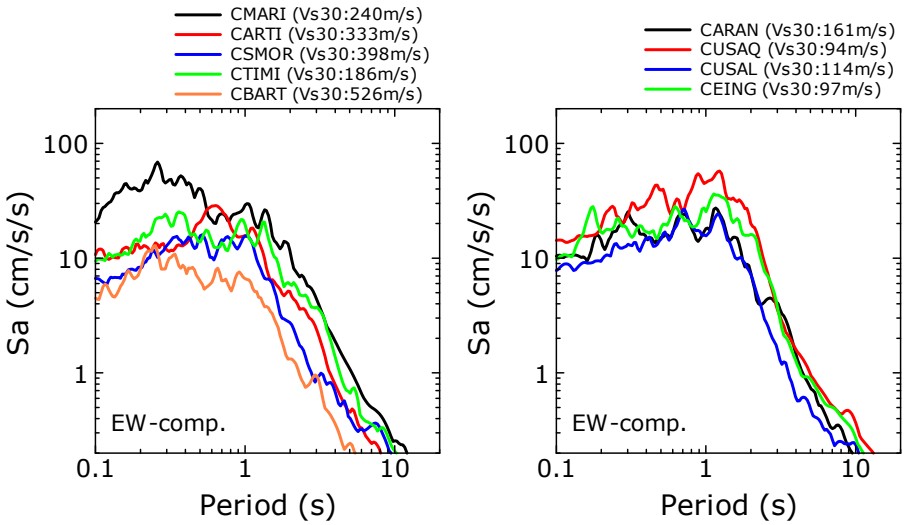

**Figure 9.** Spectral accelerations at typical sites observed in the 2019 Mesetas earthquake.

Since the observed and estimated seismic intensities were not so large and the estimated MDRs were smaller than 0.1 in all the buildings, no building loss was estimated by the proposed system in the earthquake. Whereas some damage to weak buildings was reported in the epicentral region [40], structural damage to buildings was not reported in Bogotá. The estimated building losses correspond to the actual situation.

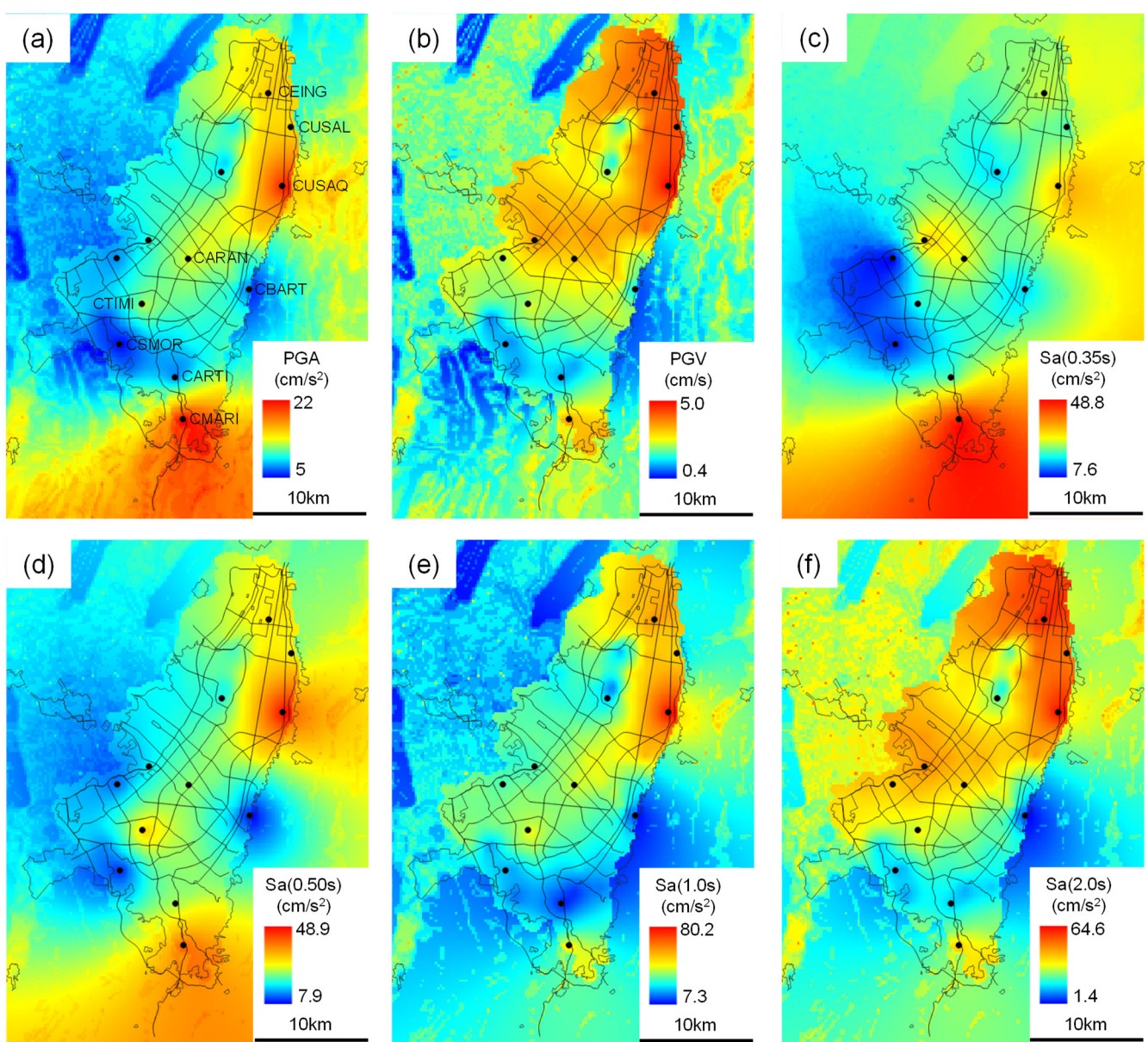

**Figure 10.** Spatial distribution of (**a**) PGA (**b**) PGV and spectral accelerations at (**c**) 0.35 s, (**d**) 0.50 s, (**e**) 1.0 s and (**f**) 2.0 s activated by the 2019 Mesetas earthquake.

## 4. Shakemaps and Building Losses for an Mw 7.0 Scenario Earthquake

As of 2023, a damaging earthquake has not been recorded in and around Bogotá since the development of the quick damage estimation system. To demonstrate economic building losses for a large earthquake, numerical simulations of seismic intensities and building losses were performed for a scenario earthquake. An Mw5.9 earthquake hit the Quetame district approximately 50 km southeast of Bogotá city and located in the Andes Mountain region in May 2008 and August 2023 (see Figure 11). In August 2023, an Mw6.0 earthquake also occurred in the active faults bounding the eastern range of the North Andes (see Figure 11). Much larger earthquakes have been anticipated in the active faults, including the Servita fault [41]. In this study, a scenario crustal earthquake with Mw7.0 corresponding to the Servita fault, an active fault located at the eastern range of North Andes close to Bogotá, as shown in Figure 11, was constructed based on active fault map information from a seismic microzonation project of Bogotá [42], as well as a methodology to estimate multi-wavelength finite fault slip distributions consistent with the

fault geometry and seismic moment of the earthquake [43]. Using the slip model obtained, the ground motions and building losses for the scenario were estimated.

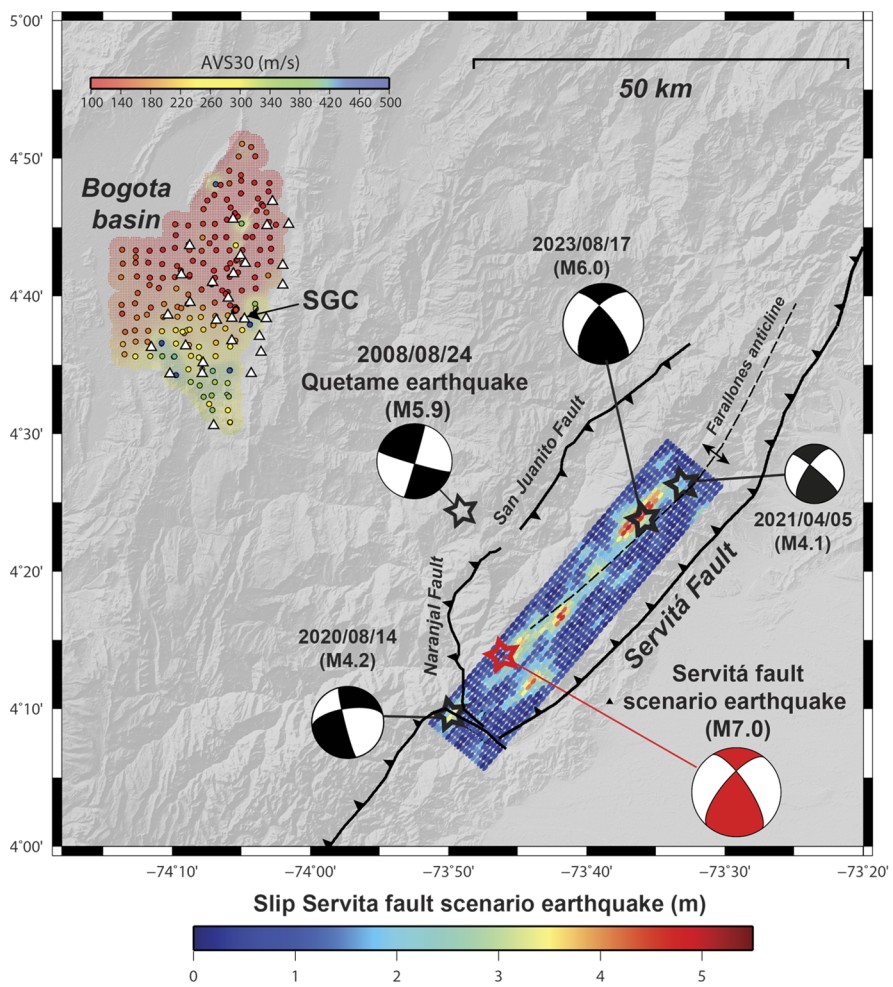

**Figure 11.** The slip model scenario of the Servita fault (Mw7.0) is shown near the southwest region of the map using blue to red color scale. The fault mechanism and epicenter of the Quetame earthquake (Mw5.8) on 24 May 2008, and of those of recent earthquakes near the Servita fault (2020, Mw4.2, 2021, Mw4.1, and 2023, Mw6.0) are displayed. The Vs30 map estimated within the Bogotá basin is shown by red to blue colored dots near the northwest corner of the map. White triangles show the strong motion stations of IDIGER and SGC in Bogotá. Trace and dip of the main active faults in the region are shown by black lines and triangles.

To perform the strong motion simulation, a hybrid simulation was used, which combines a deterministic simulation of ground motion at low frequencies (0.01–1 Hz), with a semi-stochastic simulation at high frequencies (1–30 Hz), based on a multi-scale finite fault rupture model [23,43,44]. Strong motion waveforms were calculated at all the sites of the strong motion stations by IDIGER for a bedrock outcrop condition defined as a site where Vs30 has a value of 500 m/s. This approximately corresponds to the geological condition for claystone–sandstone rock outcrops at the eastern hills of Bogotá. Distributions of strong motion indexes at the surface are obtained by multiplying amplification factors to the values of PGA, PGV, and acceleration response spectra of waveforms simulated at the bedrock by using the Vs30 map of Bogotá [23], and the empirical prediction equations for amplifications of strong motion indexes described in a previous section [24,25]. Since the shallow crustal earthquake is anticipated, surface wave amplifications were adopted to estimate surface ground motions for the scenario.

The case study presented here demonstrates the methodology's integration into simulated deterministic strong motion simulations. This integration is especially significant when prior recordings are unavailable in the seismic catalog of the region. It is worth emphasizing that the current methodology has been developed using seismic recordings whose peak accelerations are below 80 cm/s$^2$, as previously stated. However, its suitability for higher accelerations that induce nonlinear behavior necessitates further investigation. Consequently, it is imperative for future scientific research to prioritize the evaluation of these effects, enabling their full incorporation into the methodology.

Figure 12 shows the synthesized acceleration response spectra at the five stations. Since CJABO, CUSAQ, and CEING stations are located in the central part of Bogotá where the soil conditions are very soft, large amplitudes at approximately 2 s due to the surface wave site amplifications were found in the response spectra. Figure 13a–e shows the distribution of the predicted PGA, PGV, and Sa at periods of 0.35 s, 0.5 s, and 1.0 s. The PGAs and short-period spectral accelerations, such as 0.35 s and 0.5 s, show larger values in the southern part of the city because the epicentral distance is closer. The maximum PGA was estimated at 160 cm/s$^2$. On the contrary, the PGV and 1.0 s spectral accelerations show large values in the northern part since large amplifications are expected in longer period due to the very soft soil effect. The maximum PGV was estimated at 40 cm/s.

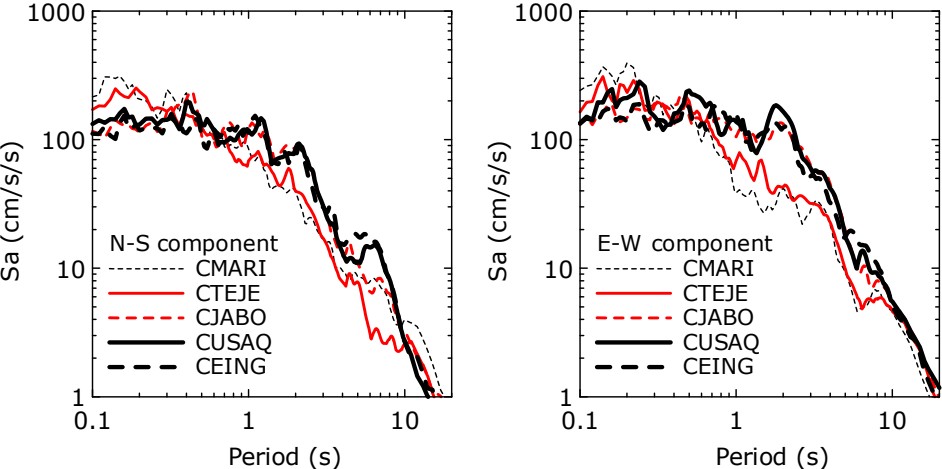

**Figure 12.** Synthesized acceleration response spectra of N-S and E-W components for the scenario earthquake at selected five stations.

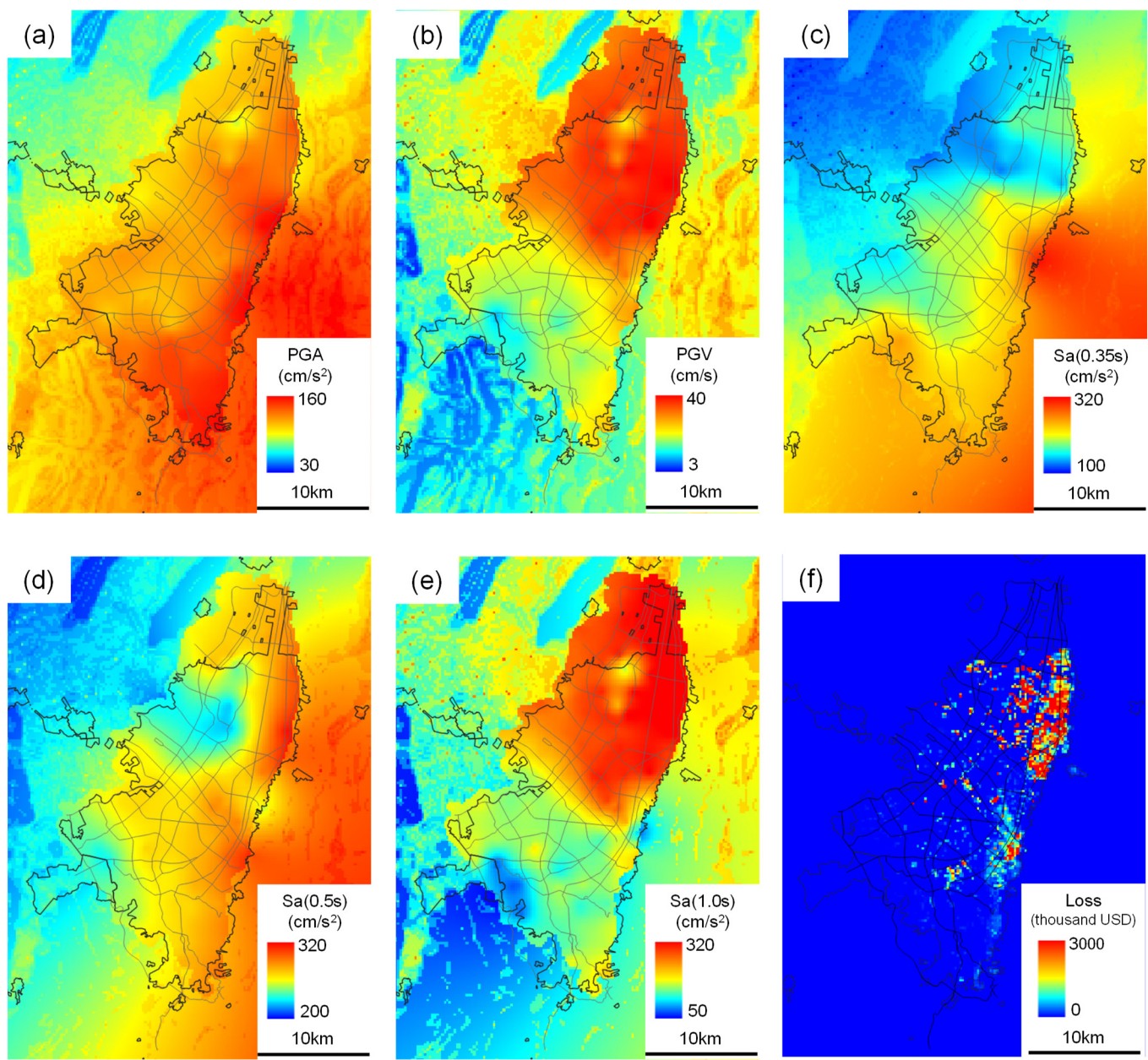

**Figure 13.** Results of seismic intensities and building losses for the scenario earthquake. (**a**) PGA, (**b**) PGV, (**c**) Spectral acceleration (Sa) at the period of 0.35 s, (**d**) Sa at the period of 0.5 s, (**e**) Sa at the period of 1.0 s, and (**f**) Building losses in thousand USD.

Figure 13f shows the distribution of the predicted building losses for the scenario. A great amount of building losses is predicted since the scale of the magnitude of the scenario is much larger than the 2019 event. Especially, heavy damage is concentrated in the northeastern part of the city because larger amplitudes of the seismic intensities were estimated due to the soft soil conditions and high-cost buildings, such as high-rise apartment houses and office buildings, are densely distributed in the area. Table 3 summarizes the predicted building loss and loss ratio for each building typology. The loss ratio in percentage is calculated from the ratio of the predicted loss to building replacement cost (i.e., using the MDR conditioned on the site spectral acceleration). The results show that the expected damages in low-rise unreinforced masonry buildings, such as ADOBE and MSC1_3, and mid- and high-rise buildings, such as PCRDMO6_12, PCRM_DMI6_12, PCRM_DMO6_12, and SC_20, are significant. The loss ratios of these typologies are

expected to be more than 10%. Especially the loss ratio of high-rise buildings such as PCRM_DMO12_20 was higher than 50%. Significant losses in low-rise buildings were caused mainly by the low seismic capacities of the unreinforced masonries. That is why they are very sensitive to ground shaking. On the contrary, a large number of losses in high-rise buildings was estimated, although the seismic capacities of the building typologies are much higher than those of low-rise buildings. As shown in Figure 13f, the losses of the high-rise buildings were concentrated in the northeastern part of Bogotá. Strong shakings in the area were predicted especially in the longer period range (1–2 s) that affect high-rise buildings due to soft soil deposits and large amplification factors of approximately 1–2 s. These results indicate that not only low-rise unreinforced buildings, but also high-rise buildings would be vulnerable in Bogotá to large crustal earthquakes.

**Table 3.** Predicted building loss and loss ratio for each building typology in the scenario earthquake.

| Structural Code | Number of Polygons (Section of Buildings) | Value (Million USD) | Loss (Million USD) | Loss Ratio (%) |
|---|---|---|---|---|
| ADOBE | 19,193 | 1358 | 259 | 19.1 |
| MSC1_3 | 1,186,283 | 69,319 | 601 | 0.9 |
| MSC4_5 | 26,913 | 4517 | 69 | 1.5 |
| MR1_3 | 206,205 | 32,196 | 0 | 0.0 |
| MR4_5 | 15,645 | 3924 | 0 | 0.0 |
| PCRDMI1_3 | 253 | 95 | 0 | 0.0 |
| PCRDMO1_3 | 14,943 | 6956 | 0 | 0.0 |
| PCRDMI4_5 | 64 | 65 | 0.3 | 0.5 |
| PCRDMO4_5 | 5190 | 3077 | 0 | 0.0 |
| PCRDMO6_12 | 3308 | 4161 | 517 | 12.4 |
| PCRM_DMI4_5 | 10,299 | 3646 | 4 | 0.1 |
| PCRM_DMO4_5 | 9904 | 5157 | 10 | 0.0 |
| PCRM_DES4_5 | 1390 | 1172 | 0 | 0.0 |
| PCRM_DMI6_12 | 588 | 585 | 97 | 16.6 |
| PCRM_DMO6_12 | 4476 | 6899 | 904 | 13.1 |
| PCRM_DES6_12 | 1365 | 2022 | 201 | 9.9 |
| PCRM_DMO12_20 | 490 | 1415 | 732 | 51.7 |
| PrFC4_5 | 12,553 | 2719 | 0 | 0.0 |
| SC6_12 | 574 | 1279 | 105 | 8.2 |
| SC_20 | 47 | 172 | 31 | 18.0 |
| PRT_CER_MUR | 11,058 | 4609 | 0 | 0.0 |
| BOD_PEQ | 42,023 | 7447 | 0 | 0.0 |
| BOD_GRN | 30,948 | 6795 | 0 | 0.0 |
| Total | 1,603,658 | 169,585 | 3520 | 2.1 |

The total amount of the predicted building losses is approximately 3.52 billion USD, corresponding to a total loss ratio of approximately 2%, as shown in Table 3. According to the report by the World Bank [45], the GDP (growth domestic product) of Colombia was about 300 billion USD in the past five years from 2016 to 2021. The predicted loss of the scenario corresponds to approximately 1% of the Colombian GDP. Moreover, attention should be paid to the fact that predicted losses cover only the direct building damage due to shaking, and the amount of losses would be larger by counting losses of infrastructures, such as roads and bridges, and indirect losses, such as business interruptions. Table 4 shows the comparison of the predicted losses with the actual economic losses of the recent worldwide M6-7 class earthquakes [46]. The table also shows the direct losses in Turkey

caused by the Turkey-Syria earthquake (Mw 7.8) on 6 February 2023 [47]. It is difficult to directly compare our estimation with the actual losses because the distance from the source and the seismic capacities of the buildings are different from each other. However, the predicted losses are almost consistent with those in previous similar scale events. Attention must be paid to the fact that building damage and losses may significantly increase in the event of a much larger earthquake near the urban areas in the Bogotá region, such as the 2023 Turkey-Syria earthquake.

**Table 4.** Comparison of estimated loss by a scenario earthquake in this study with losses of the recent worldwide inland crustal earthquakes [46,47]. The losses of the 2023 Turkey and Syria earthquake represent the losses in Turkey estimated in [47]. The losses of the other earthquakes are referred from [46].

| Earthquake | Date (YY/MM/DD) | M | Reported Loss (USD) |
|---|---|---|---|
| Haiti | 2010/1/12 | 7.0 | 7.8 billion (4.3 billion direct) |
| Canterbury, New Zealand | 2010/9/3 | 7.0 | 2.2 to 2.9 billion |
| Christchurch, New Zealand | 2011/2/22 | 6.1 | 16.5 to 25 billion |
| Christchurch, New Zealand | 2011/6/13 | 6.0 | 4.83 billion |
| Eastern Turkey | 2011/10/23 | 7.1 | 500 million to 1.0 billion |
| Turkey and Syria | 2023/2/6 | 7.8 | 18 billion (Direct loss of residential buildings)<br>9.7 billion (Direct loss of non-residential buildings)<br>6.4 billion (Direct loss of infrastructures)<br>34.2 billion (Direct loss in total) |
| Bogotá, Colombia (This study) | Scenario | 7.0 | 3.52 billion (Predicted direct building losses due to shaking) |

## 5. Conclusions

A quick estimation model of seismic intensities and ground-shaking-induced building losses for Bogotá, Colombia was developed. The model is based on the information of the strong motion observation network of Bogotá by IDIGER, the dense Vs30-based site amplifications, the building inventory, and vulnerability functions. The ground-shaking maps were developed from the observation records by a spatial interpolation technique and the Vs30 map. The economic building losses were estimated by the spectral accelerations at the period of typical building types, mean damage ratios of the vulnerability functions corresponding to the repair cost in percentage, and building replacement values in the building inventory. The shakemaps, including the distributions of PGAs, PGVs and spectral accelerations, and the building loss maps can be produced taking no longer than five minutes after receiving the seismic records from the observation sites.

The application of the system was demonstrated using the observed ground motion records during the Mesetas earthquake with Mw 6.0 in December 2019. It confirmed that larger amplitudes in the long period of approximately 1–2 s were recorded in the northeastern part of the Bogotá basin due to strong site amplifications of soft soils. In addition, the performance of the model was evaluated for a hypothetical earthquake scenario with the Mw 7.0 from the Servita fault at the eastern range of north Andes. The estimated PGAs and PGVs in Bogotá reached approximately 160 cm/s$^2$ and 40 cm/s, respectively. Furthermore, the distribution of the building damage and the total amount of the building losses were predicted. It was confirmed that the developed system allows us to quickly assess seismic intensities and shaking-induced building losses for earthquakes in real-time as well as for scenario earthquakes. The framework of the quick estimation model can be expanded to other regions and countries by implementing a seismic observation network, developing a site amplification model, and constructing the building exposure database with its corresponding fragility/vulnerability models.

It is acknowledged that the current model for the site amplifications will render better estimations on small-to-moderate earthquakes than for strong motions. However, the proposed estimation model in this study, compared to existing ones, represents a more realistic representation of the Bogotá basin given the use of dense geophysical explorations throughout the city. While better information is captured for stronger motions, future research could concentrate on utilizing additional sources of information to better represent the nonlinearities expected at the site-specific ground responses of the Bogotá basin. Likewise, future information on the impacts of strong motions in similar cities (i.e., with similar construction practices and design codes) could be used to improve the modeling inputs required for replacement cost and vulnerability functions definition. Such information contributes towards improving the accuracy of the seismic loss estimation tool. Finally, it is recognized that the proposed model outcome is portrayed as a deterministic value. Although it is found a valuable metric for communicating the expected earthquake impacts in Bogotá, future studies could include sources of uncertainty, such as the building's structural characteristics, the replacement costs, and the conditional damage ratio, among others, into the calculation procedure.

In the aftermath of disasters, national and local governments need to invest significant resources to address recovery needs. The developed model intends to help the city administrative offices and stakeholders in Bogotá to understand the significance of ground shakings and building damage immediately after an earthquake. The quick estimations of the impacts of an earthquake would lead to rapid and appropriate decision making for post-disaster responses, such as in identifying areas to be allocated resources for rescue and relief and in comprehensively estimating the budget needed for recovery and reconstruction. Furthermore, the model can be also used for considering pre-earthquake counter-measures by predicting ground shakings and building losses for anticipated future earthquakes.

**Author Contributions:** Conceptualization, Hiroyuki Miura, Masashi Matsuoka, Juan C. Reyes, Nelson Pulido, Andrea C. Riaño, Alvaro Hurtado and Raul Rincon; methodology, Hiroyuki Miura, Masashi Matsuoka, Juan C. Reyes, Nelson Pulido and Raul Rincon; software, Masashi Matsuoka and Mitsufumi Hashimoto; validation, Hiroyuki Miura, Masashi Matsuoka, Juan C. Reyes, Nelson Pulido, Andrea C. Riaño, Alvaro Hurtado and Raul Rincon; formal analysis, Hiroyuki Miura, Masashi Matsuoka, Juan C. Reyes, Nelson Pulido and Mitsufumi Hashimoto; investigation, Juan C. Reyes, Nelson Pulido, Andrea C. Riaño, Alvaro Hurtado, Raul Rincon and Helber García; resources, Juan C. Reyes, Nelson Pulido, Andrea C. Riaño, Alvaro Hurtado, Raul Rincon, Helber García and Carlos Lozano; data curation, Juan C. Reyes, Nelson Pulido, Andrea C. Riaño, Alvaro Hurtado, Raul Rincon, Helber García and Carlos Lozano; writing—original draft preparation, Hiroyuki Miura and Juan C. Reyes; writing—review and editing, Masashi Matsuoka, Juan C. Reyes, Nelson Pulido, Andrea C. Riaño, Alvaro Hurtado, Raul Rincon, Helber García and Carlos Lozano; visualization, Hiroyuki Miura, Nelson Pulido and Mitsufumi Hashimoto; supervision, Masashi Matsuoka and Juan C. Reyes; project administration, Masashi Matsuoka and Juan C. Reyes; funding acquisition, Masashi Matsuoka. All authors have read and agreed to the published version of the manuscript.

**Funding:** This study was supported in part by the Science and Technology Research Partnership for Sustainable Development (SATREPS) projects entitled "Application of State of the Art Technologies to Strengthen Research and Response to Seismic, Volcanic and Tsunami Events, and Enhance Risk Management in the republic of Colombia (Principal Investigator: Prof. Hiroyuki Kumagai)" and "The Project for Development of Integrated Expert System for Estimation and Observation of Damage Level of Infrastructure in Lima Metropolitan Area (Principal Investigator: Prof. Koichi Kusunoki)." This research was also supported by the Japan Society for the Promotion of Science (KAKENHI Grant numbers 22H01741).

**Data Availability Statement:** Earthquake source information estimated by the Red Sismológica Nacional de Colombia (RSNC) seismic network by SGC (Servicio Geológico Colombiano) were used in this study.

**Acknowledgments:** Earthquake source information estimated by the Red Sismológica Nacional de Colombia (RSNC) seismic network by SGC (Servicio Geológico Colombiano) were used in this study. This study was supported in part by the Science and Technology Research Partnership for Sustainable

Development (SATREPS) project entitled "Application of State of the Art Technologies to Strengthen Research and Response to Seismic, Volcanic and Tsunami Events, and Enhance Risk Management in the republic of Colombia (Principal Investigator: Hiroyuki Kumagai)".

**Conflicts of Interest:** Author Mitsufumi Hashimoto was employed by the company Kozo Keikaku Engineering Inc. The remaining authors declare that the research was conducted in the absence of any commercial or financial relationships that could be construed as a potential conflict of interest.

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
