# Peer review of "Quick Estimation Model for Mapping Earthquake Impacts in Bogotá, Colombia"

_ijgi, doi:10.3390/ijgi12120471_

Round 1
Reviewer 1 Report
Comments and Suggestions for Authors
The paper provides a valuable contribution in developing a quick estimation model for earthquake impacts. In the introduction, it might be beneficial to include some statistics or historical context regarding earthquake occurrences in the Bogotá region to underscore the importance of the research. The paper's readability and accessibility could be improved by breaking down complex technical details into more digestible sections with clear explanations.
Discuss how the research team handled uncertainties or data gaps in the model. What measures were taken to address these challenges?
In the conclusion, consider discussing the potential broader applications of this model for other earthquake-prone regions globally and how it can contribute to a more rapid and effective response to seismic events.
Mention the practical implications of the research for emergency response agencies, urban planners, and policymakers, highlighting how the model can inform their decision-making processes.
What specific earthquake scenarios were used to validate the model, and how well did it perform in those cases? Were there any scenarios where the model's predictions significantly deviated from actual outcomes?
Could you clarify the selection criteria for building inventory and vulnerability functions? What are the sources of this data, and how representative is it of Bogotá's building stock?
In the methodology section, it's important to explain the specific interpolation technique used to estimate ground motion intensities. How was this technique chosen, and how does it affect the accuracy of the model's predictions?
The paper mentions the need to discuss the applicability of the vulnerability-based approach for accurate building loss estimation. Could you expand on this point? What are the potential limitations or challenges associated with this approach?
In the conclusion, the paper highlights the significance of the model for city administrative offices in Bogotá. Could you elaborate on how this model could be practically integrated into their decision-making processes and disaster response plans?
Are there any plans for further research or improvements to the model, such as incorporating real-time data and monitoring, refining vulnerability functions, or expanding the model to cover a wider range of seismic scenarios?
Author Response
Our responses to the reviewer are described in the attached file. Thank you.

Reviewer 2 Report
Comments and Suggestions for Authors
The authors present an interesting study providing a quick estimation model for mapping seismic intensities and building losses for Bogotá, Colombia. The study applies many existing data (reports, databases) and combines (and upgrades) many already conducted research (majority of them are co-authorised by the authors of this study) into one comprehensive model/system for evaluation of earthquake impacts in Bogotá.
The organization of the paper is clear and systematic. The paper is well-written, and it absolutely has merit, from the scientific as well as from the practical aspect. However, before accepting the manuscript for publication in ISPRS International Journal of Geo-Information the following issues need to be addressed:
1. Line 61: The statement should be revised. The occurrence of an earthquake event of selected magnitude has its own probability, thus we cannot say that it will for sure happened.
2. Section 2.2: More explanation and justification on the selection of so low PGA level (80 cm/s2) for the determination of the site amplification factors is needed. As understood from the proposed study with the assumed PGA value the base for the presented comprehensive model (which is introduced for expected (design) earthquakes with much stronger ground motions) is established. The authors do mention this as a limitation of the study fairly, but they do not give the reasons why more real (higher) PGA was not considered in the generation phase of the proposed method. Please, justify.
3. Data mismatch: The differences in the quoted numbers of building typology classes considered in the study should be explained or eliminated: 25 classes (line 202) VS. 23 classes (Table 2).
4. Table 2: The explanation on the quoted “Structural period” values is missing. How were the values (precision 0,01 s) for each structural typology calculated (“estimated”)?
5. The applied nomenclature of the analysed building typologies (“Structural codes” – Tab. 2) is in general not associative with their terms in English language. For the readership this makes it difficult to read and follow the article. The use of more straightforward and meaningful abbreviations (codes) would be helpful.
6. To improve its readability the use of larger maps in Figure 6 would be appreciated (current maps are very small and it is hard to read the data from them).
7. In the case study No. 1 (chapter 3) it should be more exposed that the intensity of the considered earthquake was very low (max PGA (Sa) was approx. 25 (80) cm/s2 … CMARI) (see also the comment No. #2). Thus, the obtained earthquake impacts (actually no structural damage to buildings were detected) are expected. Similarly, also in the hypothetical earthquake scenario, i.e. case study No. 2 (chapter 4), the applied seismic intensity was not so large (max PGA around 160 cm/s2). The criteria for the selection of such low PGA values should be clearly quoted and justified. The dependence and accuracy of the model results obtained in the case of assumed higher PGA levels should be discussed or at least speculated. Might the results considerably change in the case of nonlinearities occurred in the model when a stronger seismic input applied?
8. Figure 13f: The scale (Loss) should be in USD (instead of COPs). Please, replace and be consistent throughout the paper.
Comments on the Quality of English LanguageEnglish language: Although this reviewer is not an expert in this area, a professional proof-reader review is warmly recommended. Some sentence structures need to be revised.
Author Response

(The authors gave the same response as above.)

Reviewer 3 Report
Comments and Suggestions for Authors
The article proposes tools for a rapid assessment of shakemap and economic losses after an earthquake in Bogota. With their new methodology, the authors propose to address some of the limitations of an existing method (LISA) in this field. The subject is interesting from both a scientific and an administrative point of view. The article is well-written, and the main elements of the methods are described. The examples proposed allow us to evaluate the capacity of the proposed method to derive shakemap and economic losses evaluation.
The reviewer suggests only minor modifications to give some precisions and limitations of the proposed methods.
p.2 - l.66: the authors should give an explicit (in meters) description of the resolution as "block level" is not a clear unit.
p.2-3: For the information regarding the potential gain of time and precision, it would have been interesting to compare with LISA. Without developing a precise comparison, the authors should at least give some global comparison as they proposed to improve this existing approach (p.2)
p.5 - section 2.2: can the authors provide the shape of the basin? From Fig. 4, there is a strong gradient in VS30 at the basin's boundaries. Is this evolution linked to the resolution chosen? How do the boundaries affect the ground motion in this area? Is there some bias in the estimation in this area that may affect the shakemap and economic losses?
p.6 - first paragraph: can the authors provide details regarding computation time for each step of the method?
p.7 - section 2.3: It seems the method proposed does not include the period of construction. From the reviewer point of view, it is a critical point, and it will strongly affect the economic loss estimation. Indeed, one may suppose design rule will improve the capacity of the structure. The same typology built before code or after code will not react in the same way. Furthermore, one may also expect damage accumulation due to moderate earthquakes and so an evolution of the period (see for instance: https://link.springer.com/article/10.1007/s10518-020-00840-0 )
One may suppose that updating the hazard map in Bogota may have a beneficial effect on structures built after this update. The authors should point out this substantial limitation in defining the vulnerability functions.
p.10 - l.250-252: What is the motivation from a methodological point of view to consider a nil value for MDRs smaller than 0.1? Is there a computational cost reduction? From a practical point of view, cracks on non-structural walls may affect other characteristics of the structures, like thermal insulation. One may also consider that repair may be done not to recover the earthquake capacity but to recover the thermal properties.
p.10- l.260: quantitative and explicit information regarding the computational time should be given. Indeed, it is one of the highlights of the authors that the method is a "Fast computation". As mentioned, a comparison with an existing approach (LISA) should be given.
Section 3: For both examples, the surface waves implication factor is supposed to be prior. In practice, how do the authors proceed, and how is the computation performed if this information is unavailable?
p13 - l.304: Is it "almost no building loss" or "no building loss"? The results should be more quantitative. As it is an existing case, a map and comparison should be given with the real data. Indeed, the authors say, "the estimated building losses correspond to the actual situation."
Some typos:
p5-l.151: the "town" of Bogota of "Since Bogota"
p.11-l278: it seems it should be a reference to Fig.4 instead of Fig.8.
p11- Fig8 right: Title EW-component (the "n" is missing)
p12-l.293: it seems some words are missing in the sentence.
Author Response

(The authors gave the same response as above.)
